# Comparative Analysis of Multiple GWAS Results Identifies Metabolic Pathways Associated with Resistance to *A. flavus* Infection and Aflatoxin Accumulation in Maize

**DOI:** 10.3390/toxins14110738

**Published:** 2022-10-28

**Authors:** Marilyn L. Warburton, Dan Jeffers, Jessie Spencer Smith, Carlos Scapim, Renan Uhdre, Adam Thrash, William Paul Williams

**Affiliations:** 1United States Department of Agriculture ARS Plant Germplasm Introduction and Testing Research Unit, Washington State University, Pullman, WA 99164, USA; 2United States Department of Agriculture ARS Corn Host Plant Resistance Research Unit, Mississippi State, MS 39762, USA; 3Department of Genetics and Plant Breeding, State University of Maringa–UEM, Maringa 59950, PR, Brazil; 4Department of Crop and Soil Science, Washington State University, Pullman, WA 99164, USA; 5Institute for Genomics, Biocomputing & Biotechnology, Mississippi State University, Mississippi State, MS 39762, USA

**Keywords:** metabolic pathway analysis, GWAS, maize (*Zea mays*), aflatoxin, *Aspergillus flavus*

## Abstract

Aflatoxins are carcinogenic secondary metabolites produced by several species of *Aspergillus*, including *Aspergillus flavus*, an important ear rot pathogen in maize. Most commercial corn hybrids are susceptible to infection by *A. flavus*, and aflatoxin contaminated grain causes economic damage to farmers. The creation of inbred lines resistant to *Aspergillus* fungal infection or the accumulation of aflatoxins would be aided by knowing the pertinent alleles and metabolites associated with resistance in corn lines. Multiple Quantitative Trait Loci (QTL) and association mapping studies have uncovered several dozen potential genes, but each with a small effect on resistance. Metabolic pathway analysis, using the Pathway Association Study Tool (PAST), was performed on aflatoxin accumulation resistance using data from four Genome-wide Association Studies (GWAS). The present research compares the outputs of these pathway analyses and seeks common metabolic mechanisms underlying each. Genes, pathways, metabolites, and mechanisms highlighted here can contribute to improving phenotypic selection of resistant lines via measurement of more specific and highly heritable resistance-related traits and genetic gain via marker assisted or genomic selection with multiple SNPs linked to resistance-related pathways.

## 1. Introduction

Aflatoxins are carcinogenic secondary metabolites secreted by some species of *Aspergillus* fungi. Although aflatoxin production has been reported in as many as 17 *Aspergillus* species [1], the most economically important, and therefore the most studied, aflatoxin producers are *A. flavus* and *A. parasiticus* [2,3]. *Aspergillus flavus* and *A. parasiticus* can colonize and contaminate almost any crop during storage, but they are also pre-harvest pathogens of oilseed crops, including maize [4,5]. Aflatoxin contamination of maize grain caused by pre-harvest infection, mainly by *A. flavus,* occurs worldwide. [6,7]. Most commercial maize hybrids are susceptible to infection, which causes Aspergillus ear rot (AER), significantly reduces the value of the grain, and poses health risks to humans and animals. When consumed, aflatoxins cause liver dysfunction, teratogenic defects, and immunosuppression [8,9,10,11]. Strains of *A. flavus* behave differently on different growth substrates, including growth rates, timing of reproduction, and level of aflatoxin production. These differences suggest that the production of primary and secondary metabolites depends on the food source of the fungus, and may hint at underlying cellular processes, in both the fungus and the infected plant [12,13,14,15].

In maize, host-plant resistance to aflatoxin accumulation is a quantitative trait with relatively low heritability that interacts significantly with environmental effects [16]. Susceptibility is increased by heat and drought stress, as stressed plants are more susceptible to pathogens in general, but to *A. flavus* infection, growth, and aflatoxin production in particular [17]. A few studies have worked to identify chromosomal regions associated with resistance, including larger regions (QTL) [16,18,19] and smaller genomic regions via associative mapping. Mapping these fungal resistance QTL may be more useful when looking within tropical maize genotypes, which have evolved under environmental conditions favoring fungal growth, and which may have promoted the evolution of resistance [20,21].

Association mapping (particularly genome-wide association study, GWAS) is a powerful tool to map chromosomal regions associated with traits of interest, and to identify natural variation and genes related to pathogenicity in animals [22,23], humans [24,25], and crops, including maize [26] and specifically resistance to ear rot by *A. flavus* [20,21]. One drawback of GWAS is low statistical power when studies include relatively few samples, e.g., when phenotyping costs are prohibitive. In addition, when genes exert small effects on complex polygenic traits, they may not be identified, especially if the value of the association effect is also influenced by the environment [27,28].

Another way to obtain information about the genetic component of resistance to aflatoxin accumulation is through analysis of metabolic pathways. These studies focus on the combined effects of many genes that are grouped according to their shared biological function [29,30]. Analysis of these groupings allows interpretation of gene function and provides interrelated information for cellular functions [31,32,33,34,35,36,37]. A pathway analysis has been used in studies of aflatoxin accumulation in maize [21,30]. Tools such as the pathway association study tool (PAST) have been developed to help identify pathways associated with an increase in the phenotypic expression of a trait [27,30]. Particularly for plant species, PAST is a quick and easy-to-use tool to exploit GWAS data for analysis of metabolic pathways to identify genes and epistatic interactions between genes associated with increased or decreased phenotypic expression of a trait of interest [38,39].

The first use of the PAST protocol identified metabolic pathways in maize plants responding to inoculation with *A. flavus* spores [30] and observed that the jasmonic acid (JA) biosynthesis pathway was important in the defense of maize inbred lines against the fungus. The authors observed that there was at least one allelic variant for each step of the JA biosynthesis pathway that conferred an incremental decrease in aflatoxin level observed among the inbred lines in the panel. Other studies found that JA also increases resistance to necrotrophic pathogens [40,41] and resistance against herbivore attacks [42]. Various metabolic compounds and proteins have been seen to increase in expression or accumulation, or increase resistance responses, in plants exposed to fungi. For example, phytoalexin hormones [43,44] may be used by the plant to reduce initial infection, reduce spread of fungal infection between cells, or reduce production of mycotoxins. Provitamin A carotenoids, particularly beta-carotene and beta-cryptoxanthin were associated with decreased aflatoxin in 120 maize hybrids, likely due to antioxidant activity that limits the production of aflatoxin [45]. Studies of *A. flavus* infection and aflatoxin production in maize also point to important roles for some proteins, genes, and metabolites. Genetic and association mapping found genes that produce enzymes that break down chitin to be linked to reduced aflatoxin in maize; these enzymes can defend against fungi, as chitin is found in the cell walls of fungi, yeast, and algae, as well as the exoskeleton of insects [46]. The production of phenol-like compounds in cuticular waxes has also been found in *A. flavus* resistant maize lines [47]). Other genes and proteins that have been suggested to provide protection to maize against *A. flavus* infection and aflatoxin accumulation include a trypsin inhibitor [48], α-amylase [49], a pathogen related protein PR10 [50], and lipoxygenase genes [51,52,53].

Considering the immense global health burden and economic importance of damage caused by aflatoxin, studying this trait with tools that provide an accurate mapping of related, small-effect genes that in combination provide a resistance phenotype is essential. Knowledge gained from such work will facilitate marker-assisted selection and allow the creation of an ideotype of a resistant plant, which breeders could incorporate into genomic selection or phenotypic selection programs. Linking GWAS with the study of metabolic pathways can provide more information on genetic mechanisms and the biological relevance of important traits [38,54]. Importantly, since more than one GWAS dataset has been created for the study of aflatoxin accumulation resistance in maize, we can now compare the outcome of pathway analyses on each and seek common mechanisms that may help breeders in the future focus their resistance improvement efforts. The present work used the PAST tool to analyze four different maize GWAS datasets to identify metabolic pathways and associated genes contributing to the reduction of aflatoxin accumulation in maize, thereby contributing to greater understanding of the genetic architecture and the physiological mechanisms that affect the maize phenotype.

## 2. Results & Discussion

The 197 pathways associated with increasing and decreasing expression of the traits in the four studies at a *p* level ≤ 0.05 can be found in Appendix A. Comparing the pathways in common between studies (either different datasets, or the increasing vs. decreasing analyses) is not as straightforward as hoped, because some are named differently in different annotated versions of the maize genome; multiple pathways may create the exact same metabolite (and many create related metabolites); and some named pathways overlap others. However, there were 34 pathways with identical ID numbers or ID names identified in more than one study, and 64 that were matching based on metabolite produced (Figure 1; Appendix A). Four pathways (PWY-5410 (traumatin/LOX biosynthesis); PWY1F-467 (also known as R-ZMA-1119316.1, phenylpropanoid biosynthesis); THRESYN-PWY (threonine biosynthesis); and CHLOROPHYLL-SYN (chlorophyll synthesis) were identified simultaneously in 3 panels. This large number of commonalities may indicate that the PAST program is finding common genetic mechanisms between different populations. Different GWAS analyses of the same quantitative trait rarely find the same SNP loci significantly associated. The strict significance thresholds used in GWAS cause much of the data to be discarded, many of which will be false negative signals. These SNPs will contribute to raising the significance of association of a pathway in the PAST analysis, instead of being discarded, and may be why so many common pathways were identified in the current study. However, in the current study, more common pathways were identified in panels that had used the same version of the reference genome sequence to identify SNPs (Brazil and CHPRRU1) than panels with nearly the same entries and phenotyping (CPHRRU1 vs. 2). This is also a problem comparing different GWAS outputs and many common SNPs and pathways may not be identified because of variant calling differences.

Just over half of the associated pathways (~100 of 193) were involved in a limited number of plant functions (descriptions of plant functions can be found in the notes in Appendix A). Looking at these related and duplicated pathways, it is evident that there are some common mechanisms of resistance to *A. flavus* in maize. They are described below according to their annotated functions, and a synthesis of the pathways fond within each function can be found in Table 1. The numbers of pathways in the sections below may not correspond precisely to the numbers of pathways in Table 1 because several of the pathways were found (and thus counted) multiple times in the analyses but are only listed once in the table.

### 2.1. Plant Signaling

Twenty-one pathways in increasing and decreasing aflatoxin analyses were related to plant signaling via terpenoid metabolites. These pathways include the biosynthesis of the brassinosteroids (BR) class of steroid hormones that regulate plant development and physiology. Brassinosteroids and salicylic acid signaling pathways and genes mediate local and systemic activation of defenses following fungal infection of maize silks [55] and seedlings [56]. Brassinosteroids turn on BR responsive Transcription Factors, which control cell elongation, growth and tissue differentiation; and upregulate gibberellin (GA) and auxin pathways, which were also identified in our common pathway analyses. Gibberellins are diterpenes involved in plant development. In addition, three associated pathways synthesize or contribute to production of jasmonic acid and five others produce known signaling molecules. For example, the ASC-GSH pathway is important for abiotic and biotic stress response for defense against ROS with higher ASC constitutive levels in an ear rot resistant versus susceptible inbred [57] and higher ascorbate peroxidase (APX) in the rachis of AER resistant lines [58]. Examples of AER resistant vs. susceptible maize lines can be found in Figure 2. The jasmonate biosynthetic pathway is a key determinant for resistance to AER [52,59].

Terpenes and terpenoids are the more commonly studied phytoalexins, which are small secondary metabolites that are induced following attack by multiple pathogenic organisms including insects and fungi [43,44]. Another diterpene identified in this multi-pathway analysis in addition to the phytoalexin diterpenes is ent-kaurene, which represents the first committed step in the biosynthesis of gibberellins in the terpenoid biosynthesis pathway. ent-kaurane–related diterpenoids are collectively called kauralexins, and the production of kauralexins is induced by insect and fungal attack. This production is influenced by a combination of the synergistic plant hormones jasmonic acid and ethylene [41]. Terpenes are synthesized from the basic five-carbon isoprene unit (C5H8) by the mevalonate pathway. The isoprene units are added together via condensation reactions to form branched and cyclized isoprene polymers (hemiterpenes, monoterpenes, sesquiterpenes, diterpenes, sesterterpenes, triterpenes, tetraterpenes and polyterpenes). Several of these terpenes and terpernoids are associated with plant-fungal interactions [60]. Triterpenes, tetraterpenoids (including carotenoids and ABA) and sesquiterpenes were also identified in our study. It is evident that the production of these inducible defense and signaling molecules play a large role in maize’s response to infection by *A. flavus*.

### 2.2. Structural Components

Fifteen pathways associated with the production of phenylpropanoids, including phenolics and anthocyanins, were found in both the increasing and decreasing analyses. Phenylpropanoids make lignin and cell walls, and many phenolic acids are components of the cell wall. Following infection, decreases in soluble phenylpropanoids occur with an increase in cell wall bound pheylpropanoid metabolites [61]. Gembeh et al. [47] found a higher percentage of phenol-like compounds in wax from a resistant maize line (GT-MAS: gk) than in waxes from the susceptible lines. O-Methylated compounds in grasses can serve as precursors of lignin for the cell wall, and O-Methylated flavonoids function as antimicrobial phytoalexins [62]. In maize, fungal inducible O-Methylated flavonoids form xilonenin with antifungal activity against *Fusarium verticillioides* and *F. graminearum*, as well as other antifungal activity compounds [63]. Flavonoids may also play an important role in resistance response in other signaling pathways leading to the production of reactive oxygen scavengers [64]. Increased levels of flavonoids were found in the resistant inbred TZAR102 versus the susceptible inbred Va35 and following *A. flavus* infection flavonoids modulated toxin accumulation [65].

Eighteen pathways involved in the production of other cell wall components were also prevalent in the pathway analyses. Specifically, 9 pathways involved in the production of cellulose, hemicellulose, lignin, glycoprotein and suberin, were identified. Induced cell wall deposition of phenolics increase resistance to fungal penetration [66], and so in addition to the anti-fungal activity noted above, a stronger physical barrier could preclude fungal infection and spread. Callose deposition induced by pathogen associated molecular pattern, (PAMP) is regulated by extracellular DIMBOA [67]. Induced systemic resistance to *Fusarium verticillioides* down-regulated pathways for cellulose production while up-regulating genes involved in reinforcement of the cell walls in a previous study [55]. The production of the three aromatic amino acids, L-tryptophan, L-tyrosine, and L-phenylalanine, derived from the common precursor chorismate, have been reported to increase production of cell walls and phenolic compounds, particularly after fungal attack; 4 additional pathways that create chorismate or these amino acids were identified in the current study.

Nine pathways involved in the production of cell membranes, especially choline and sphingolipid biosynthesis were associated with increased and decreased resistance in the pathway analyses. Cell membranes are known to function in plant defense because they carry sets of pattern-recognition receptors (PRRs) that detect cellular components that do not come from the plant itself, often shed by or part of invading pathogens. These PRRs recognize microbe-associated molecular patterns (MAMPs) and activate pattern-triggered immunity (PTI) when they are triggered. However, these PRRs are not the cell membrane itself, and not the subject of the pathways identified in our study. Choline is a cell membrane associated metabolite necessary for plant development and is an antioxidant in many plant tissues [68]. It is known to increase resistance to abiotic stresses by decreasing ROS accumulation and membrane lipid peroxidation, which disrupts the structure and function of cell membranes under stress. Although choline has not been identified as a compound associated with resistance to pathogens in many plant studies, one found an increase in choline levels following infection by bacteria [69], possibly in response to damage caused by the bacteria. Sphingolipids are structural components of the plant cell membranes that increase fluidity and biophysical order, and function as signaling molecules in plant stress responses [70]. Although also generally involved in abiotic stress responses, sphingolipids have been associated with resistance to bacteria and fungi in Arabidopsis [71].

### 2.3. Defense Compounds

Many compounds are produced specifically to defend against fungi. In our analysis, 18 defense compound pathways were identified. In addition to the phenolics and polyphenols which form part of the cell wall, compounds identified in the current study are created by 7 pathways relating to hydroxycinnamic acids, including chlorogenic acid. In maize, ferulic acid and p-coumaric acid are the most abundant in the hydroxycinnamic complex [72,73], and have been shown to enhance resistance to biotic and abiotic stresses during the growing season and grain storage [74,75]. This enhancement is probably due to the extensive network of cross-links created by dimerization and trimerization that stop the herbivory of insects in stored grains and prevents the penetration of pathogens during the growing season [76,77,78]. The biosynthesis of non-specific defense compounds including canavanine, citrulline, L-Nδ-acetylornithine, phenylethanol, salicylate and dhurrin, were each identified once, and acetaldehyde twice, in the current study.

Lipoxygenases are an important molecular family distributed in plants and can be classified into two large subfamilies of 9-lipoxygenase (9-LOX) and 13-lipoxygenase (13-LOX) [79]. Increased levels of LOX activity have been reported in various plant-pathogen systems, including maize [80]. Compounds created by the lipoxygenases and 9-divinyl ether synthase (9-DES) in response to fungal challenges include oxylipins (12-OPDA by the action of 13-LOXs, and 10-OPDA by the action of 9-LOXs) that then contribute to the production of jasmonic acid, green leafy volatiles [81,82], and specific compounds such as traumatin, divinyl ether, and others. 9-LOX and 13-LOX related pathways were both identified 3 times in the current study. Some of the proteins or genes identified in previous studies but not in the current study include the trypsin inhibitor gene, α-amylase enzyme, and PR10 genes studied by Chen et al. [48,50] and Woloshuk et al. [49]. These genes are not part of a specific, known pathway or mechanism and thus would not have been identified by the PAST analysis.

### 2.4. Other Pathways

Several other pathways were identified in analysis of the four datasets with PAST. Six pathways were identified that directly or indirectly synthesize coenzyme A, and 12 others synthesized vitamins (including pro-A, B, C and E). Many of the remaining 92 pathways identified in this analysis and listed in Appendix A include basic cell functions needed for basic life and growth functions, including the production of amino acids (31 of the remaining pathways); nucleotide biosynthesis and conversion (10 pathways); starch, sugar and fatty acid metabolism (19 pathways); the nitrogen cycle (5 or more pathways); and the carbon cycle (7 or more pathways), among others. Pathway groupings here are not mutually exclusive, as most of the biological functions of a cell overlap and lead from one to another and back in cyclical processes. However, what is apparent is that the pathways mentioned here do not lead specifically to a single resistance compound or mechanism, and most likely illustrate that strong, actively growing plants are more resistant to pathogens in general and can more easily contain an early infection of *A. flavus)*, than can weak or degrading plants and tissues. *A. flavus* is a decomposing fungus, and rarely causes extensive damage to an actively growing plant. There is also evidence that aflatoxin production increases after a plant experiences stress [83] and thus, the association of growth and maintenance pathways with reduced aflatoxin in multiple analyses of this study is not surprising.

## 3. Conclusions

These results can guide research to identify phenotyping targets useful for aflatoxin resistance. Changing the levels of production of plant hormones, plant signaling, and probably pathogen recognition and membrane function are somewhat obvious but may be hard to implement as resistance mechanisms because they are very complicated responses and interfere with each other. However, specific defense compounds, (terpenoids, phenylpropanoids, hydroxycinnamic acids, including specifically chlorogenic acid) may be specifically measured and selected for in future programs to increase resistance to aflatoxin accumulation in maize. Stronger cell walls including increased phenolic compounds will almost certainly lead to reduced fungal infection and reduced aflatoxin levels, but negative nutritional properties of such mechanisms may need to be investigated. In addition to identifying possible useful targets of plant improvement for aflatoxin reduction, it is apparent in this study that PAST analysis finds pathways in common between independent GWAS studies, which may help to illuminate common genetic mechanisms contributing to a trait of interest in different populations of the same species.

## 4. Materials and Methods

Four GWAS studies were included in the comparative analysis of aflatoxin accumulation resistance in maize. One study of 282 temperate and semi-tropical inbred maize lines was characterized by the CHPRRU twice independently via genotyping by sequencing (GBS), leading to two separate genotypic data sets ([31] and Appendix A). The same phenotypic data, consisting of aflatoxin concentrations measured from replicated field trials of lines inoculated with *A. flavus* (from [31]) were used for GWAS with both GBS data sets (CHPRRU1 and CHPRRU2). A third data set (BRAZIL) came from a panel of 320 tropical field corn and popcorn inbred lines grown in replicated field trials and that had been inoculated with *A. flavus* and scored for Aspergillus ear rot (AER). The lines were also genotyped by sequencing and presented in Bengosi Bertagna et al. [21]. A fourth data set (CIMMYT) of 393 semi-tropical inbred lines were genotyped by sequencing and aflatoxin concentrations were measured from replicated field trials of lines inoculated with *A. flavus*; data were presented in Suwarno et al., (in review).

The output of Mixed Linear Model (MLM) GWAS from the four previously analyzed studies were obtained and run through the Pathway Analysis Study Tool [27] as per Tang et al. [30]. For all four studies, the entire data sets created with GWAS using the Mixed Linear Model in TASSEL [84] were input into the PAST tool run from the MaizeGDB website (https://maizegdb.org/past; accessed on 27 June 2022). The data used in PAST included the following: SNP-trait association values for significance (p); correlation (R2 or proportion of the phenotypic variation accounted for); effect values along with the calculated LD values for D’ and R2; and p between each marker SNP and its closest neighboring SNPs (50 upstream and 50 downstream). The association and effects files were uploaded without filtering for significance.

SNPs were then assigned to genes by seeking a tagSNP for each block of linked SNPs as outlined in Tang et al. [30]. Briefly, a decision tree is programed into PAST to determine which SNP to call the tagSNP. If LD analysis showed no linkage between a SNP and any other SNPs, that SNP becomes the tagSNP. If there is linkage between a SNP and a block of other SNPs (upstream or downstream), selection of the tagSNP from any of the linked SNPs is chosen as the SNP with the largest positive or negative trait association effect value, lowest *p* value, and shortest distance to all other SNPs in the linkage block. Choosing the tagSNP with the greatest effects on the trait under analysis reduces dimensionality of the dataset. The gene(s) causing the SNP-trait association was then assumed to be within 1 Kb of the SNP. When a gene was located, the SNP-trait association effect value was transferred to the gene and used for the gene-set enrichment analysis as per Tang et al. [30]. SNPs were assigned to genes using a window of 1Kb around each SNP and an R^2^ of 0.8 to determine linkage between SNPs.

Pathways associated with both increasing and decreasing expression of the trait (levels of aflatoxin) were identified. Only pathways with at least 5 annotated genes were analyzed to avoid small sample size effects. Significance of association between pathways and the phenotypes was determined by creating 1000 random distributions of gene effects and comparing the actual effects to randomly sampled effects. The outputs (8 files total; pathway analyses associated with increasing and with decreasing levels of aflatoxin from the CHPRRU1, CHPRRU2, BRAZIL, and CIMMYT datasets), were compared to find pathways in common between them. Pathways with the same name or pathway ID in the CornCyc database (https://maizegdb.org/metabolic_pathways; accessed on 27 June 2022) used by PAST online, were considered a perfect match. In addition, pathways that lead to the same end product were also considered a match, as there are often two or three pathways (usually with most of the same genes) that can synthesize the same compound in the same organism. Finally, pathways leading to related compounds or biological functions were also noted and discussed.

## Figures and Tables

**Figure 1 toxins-14-00738-f001:**
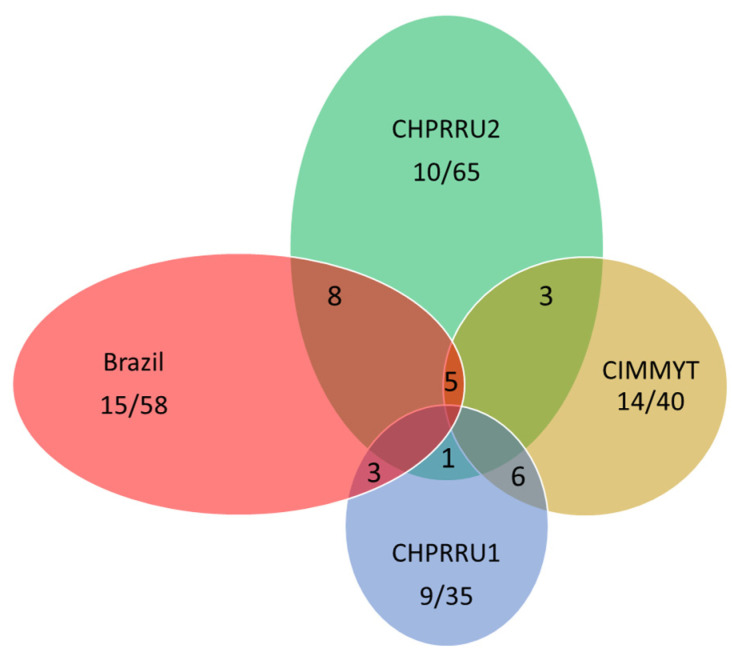
Overlapping pathways identified via PAST in different GWAS panels (Brazil, CHPRRU1, CHPRRU2, and CIMMYT). In four different panels, size of the bubble corresponds to the number of pathways identified as associated with aflatoxin level at *p* < 0.05 in each; this is also indicated in the numbers within the bubble after the slash. Number of pathways identified in at least one other panel is shown before the slash. Pairwise comparisons are the numbers in the intersections (only taken 2 panels at a time, or all at once, center number).

**Figure 2 toxins-14-00738-f002:**
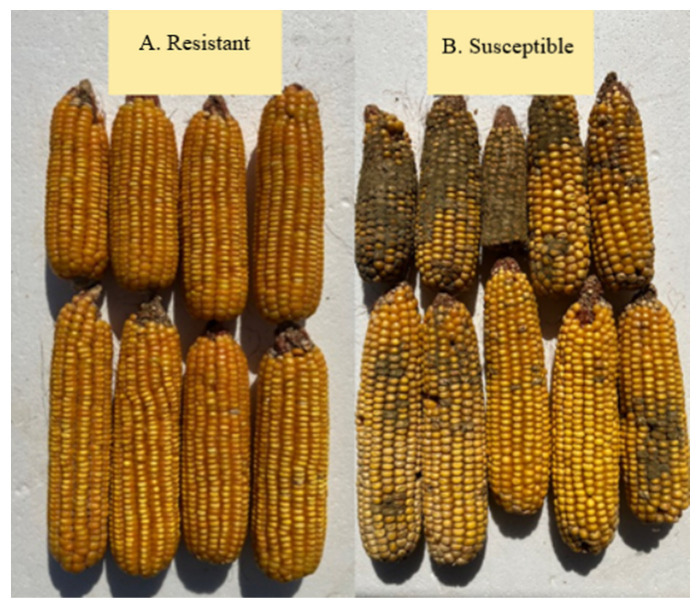
Examples of Aflatoxin Ear Rot. (**A**) resistant vs. (**B**) susceptible maize hybrids.

**Table 1 toxins-14-00738-t001:** Pathways were identified that contributed to four major mechanisms of resistance to aflatoxin accumulation in maize, including plant signaling, the biosynthesis of structural components, the biosynthesis of defense compounds, and the biosynthesis of other metabolites, primarily, Co-enzyme A and vitamins. These categories may contain pathways that appear in other categories as well, as they are not mutually exclusive. A full list of pathways and notes on the function of each can be found in Appendix A.

Plant Signaling		Structural Components		Defense Compounds		Other Metabolites	
PW-ID	Pathway Name	PW-ID	Pathway Name	PW-ID	Pathway Name	PW-ID	Pathway Name
R-ZMA-1119618.1	13-LOX and 13-HPL PW	** *Phenylpropanoids* **		PWY-5409	divinyl ether biosyn. II (13-LOX)	** *Coenzyme A* **	
R-ZMA-5608118.1	auxin signalling	PWY1F-467	phenylpropanoid biosyn., initial rxns	PWY-5410	traumatin and (Z)-3-hexen-1-yl acetate biosyn.	COA-PWY	coenzyme A biosyn. I
R-ZMA-1119456.1	brassinosteroid biosyn. II	PWY1F-FLAVSYN	flavonoid biosyn.	PWY-5751	phenylethanol biosyn.	PWY-4221-1	superPW of pantothenate and coenzyme A biosyn. II
PWY-2981	diterpene phytoalexins precursors biosyn.	PWY-3181	L-tryptophan degrad. VI	PWY-6040	chlorogenic acid biosyn. II	COA-PWY-1	coenzyme A biosyn.
PWY-5409	divinyl ether biosyn. II (13-LOX)	PWY-361	phenylpropanoid biosyn.	PWY-6330	acetaldehyde biosyn. II	LIPAS-PWY	triacylglycerol degrad.
R-ZMA-1119566.1	divinyl ether biosyn. II (13-LOX)	PWY-5059	pinobanksin	PWY-6333	acetaldehyde biosyn. I	PWY-5046	2-oxoisovalerate decarbox. to isobutanoyl-CoA
PWY-5032	ent-kaurene biosyn. I	PWY-5160	rose anthocyanin biosyn. I	PWY-6673	caffeoylglucarate biosyn.	** *Vitamin biosynthesis* **	
PWY-5120	geranylgeranyldiphosphate biosyn.	PWY-5313	superPW of anthocyanin biosyn.	PWY-6922	L-Nδ-acetylornithine biosyn.	ARO-PWY	chorismate biosyn. I
PWY-5035	gibberellin biosyn. III	PWY-5868	simplecoumarins biosyn.	PWY-861	dhurrin biosyn.	CAROTENOID-PWY	superPW of carotenoid biosyn.
PWY-102	gibberellin inactivation I (2β-hydroxylation)	PWY-641	proanthocyanidins biosyn. from flavanols	R-ZMA-1119261.1	salicylate biosyn.	FOLSYN-PWY-1	superPW of tetrahydrofolate biosyn.
R-ZMA-5679411.1	gibberellin signaling	PWY-6435	4-hydroxybenzoate biosyn. V	R-ZMA-1119344.1	hydroxycinnamic acid serotonin amides biosyn.	PWY-1422	vitamin E biosyn. (tocopherols)
R-ZMA-1119486.1	indole-3-acetate biosyn. I	PWY-6457	trans-cinnamoyl-CoA biosyn.	R-ZMA-1119444.1	canavanine biosyn.	PWY-3841	folate transformations II
PWY-581	indole-3-acetate biosyn. II	PWY-6673	caffeoylglucarate biosyn.	R-ZMA-1119495.1	citrulline biosyn.	PWY-4221-1	superPW of pantothenate and CoA biosyn. II
PWY-6219	indole-3-acetate inactivation VIII	PWY-6787	flavonoid biosyn.	R-ZMA-1119566.1	divinyl ether biosyn. II (13-LOX)	PWY-5944	zeaxanthin biosyn.
PWY-735	jasmonic acid biosyn.	R-ZMA-1119316.1	Phenylpropanoid biosyn.	R-ZMA-1119618.1	13-LOX and 13-HPL PW	PWY-5947	lutein biosyn.
PWY-6220	jasmonoyl-amino acid conjugates biosyn. I	R-ZMA-1119582.1	Phenylpropanoid biosyn., initial rxns			PWY-882	ascorbate biosyn. I
PWY-6233	jasmonoyl-amino acid conjugates biosyn. II	** *Other Cell Wall Components* **				PWYBWI-7081	carotenoid biosyn. (from lycopene)
NONMEVIPP-PWY	methylerythritol phosphate PW	PWY-1001	cellulose biosyn.			R-ZMA-1119309.1	aminopropanol biosyn.
R-ZMA-1119615.1	Mevalonate PW	PWY-1121	suberin monomers biosyn.			PWY-5188	tetrapyrrole biosyn. I
PWY-5805-ARA	nonaprenyl diphosphate biosyn. III	PWY-3181	L-tryptophan degrad. VI				
R-ZMA-1119261.1	salicylate biosyn.	PWY-4821	UDP-α-D-xylose biosyn.				
R-ZMA-1119438.1	Secologanin and strictosidine biosyn.	PWY-5659	GDP-mannose biosyn.				
PWY-5203	soybean saponin I biosyn.	PWY-7120	esterified suberin biosyn.				
PWY-5121	superPW of geranylgeranyldiphosphate biosyn. II (MEP)	PWY-7343	UDP-α-D-glucose biosyn. I				
PWY-5053	superPW of gibberellin GA12 biosyn.	R-ZMA-1119316.1	Phenylpropanoid biosyn.				
PWY-5410	traumatin and (Z)-3-hexen-1-yl acetate biosyn.	R-ZMA-1119582.1	Phenylpropanoid biosyn., initial rxns.				
PWY-6275	β-caryophyllene biosyn.	R-ZMA-5655101.1	xyloglucan biosyn.				
		PWY-3461	tyrosine biosyn. II				
		PWY-3481	superPW of phenylalanine and tyrosine biosyn.				
		ALANINE-DEG3-PWY	L-alanine degrad. III				
		ALANINE-SYN2-PWY	L-alanine biosyn. II				
		ARO-PWY	chorismate biosyn. I				
		PWY-6629	superPW of tryptophan biosyn.				
		*Cell Membrane Components*					
		PWY-3561	choline biosyn. III				
		PWY4FS-5	superPW of phosphatidylcholine biosyn.				
		PWY-5129	sphingolipid biosyn.				
		R-ZMA-1119276.1	choline biosyn. III				

## Data Availability

Previously published data used in this study are cited; the new data presented in this study are available in Appendix A (https://doi.org/10.5281/zenodo.7026681, accessed on 26 August 2022).

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
