# Peer review of "Comparative Analysis of Multiple GWAS Results Identifies Metabolic Pathways Associated with Resistance to A. flavus Infection and Aflatoxin Accumulation in Maize"

_toxins, 2022, doi:10.3390/toxins14110738_

Round 1

Reviewer 1 Report

The article is original and interesting. I suggest to be checked by a native English speaker for grammar/ spelling mistakes.

Then add one or few suggestive Figures, an synthetic Table to be included in the article

Author Response

Reviewer 1:

The article is original and interesting. I suggest to be checked by a native English speaker for grammar/ spelling mistakes.

It has been checked by four native English speakers now. Thank you.

Then add one or few suggestive Figures, an synthetic Table to be included in the article

We have added table 1 on page 5 and figure 2 on page 8. We hope these will do. There are several supplemental tables. I have to ask if Table 1 is really what the reviewer and editor had in mind, however, because although it is as short as I could make it, as a “synthetic table” it couldn’t be that short and it looks very long for the print article.

Reviewer 2 Report

Dear authors: 

The manuscript brings a relevant review on the resistance to A. flavus infection and aflatoxin accumulation in maize. It is also well written is within the Toxins norms. 

I suggest to accept the manuscript after making a few corrections, as follows: 

- List authors and affiliations; 

- Material and methods: it’s short, it has only 3 paragraphs. I suggest describe in more details so it can be reproduced; 

- Conclusion: please, rewrite the conclusion, be more concise. The lines 270-278 are part of the discussion of the paper, where the results found can help in future research, they are not elucidating the author's conclusion about the work done. 

Best regards. 

Author Response

Dear authors: 

The manuscript brings a relevant review on the resistance to A. flavus infection and aflatoxin accumulation in maize. It is also well written is within the Toxins norms. 

I suggest to accept the manuscript after making a few corrections, as follows: 

- List authors and affiliations; 

We have added the authors and affiliations. I don’t know how it was cut from our template, but thank you to the reviewer for catching that!

- Material and methods: it’s short, it has only 3 paragraphs. I suggest describe in more details so it can be reproduced; 

All the data were generated from other published articles, so we didn’t describe how that data was generated, but we have added more to the paragraph on the pathway analysis used in the current study. The expanded paragraphs are now in lines 325 – 343.

- Conclusion: please, rewrite the conclusion, be more concise. The lines 270-278 are part of the discussion of the paper, where the results found can help in future research, they are not elucidating the author's conclusion about the work done. 

I have added a little to clarify precisely where these results can help in future breeding efforts of aflatoxin accumulation resistance in maize. I could not shorten this section because reviewer 3 asked me to make it longer, so a balance must be kept.

Best regards. 

Thank you. The same to you.

Reviewer 3 Report

Manuscript title: Comparative analysis of multiple GWAS results identifies metabolic pathways associated with resistance to A. flavus infection and aflatoxin accumulation in maize

The manuscript is a good attempt to gather information about metabolic pathways associated with resistance to A. flavus infection and aflatoxin accumulation in maize.

1. The conclusion is written so brief, it should be improved and some future prospects must be included

Author Response

The manuscript is a good attempt to gather information about metabolic pathways associated with resistance to A. flavus infection and aflatoxin accumulation in maize.

  1. The conclusion is written so brief, it should be improved and some future prospects must be included

I have added a little to clarify precisely where these results can help in future breeding efforts of aflatoxin accumulation resistance in maize. I could not lengthen this section more than the current paragraph because reviewer 2 asked me to make it shorter, so a balance must be kept.

Round 2

Reviewer 1 Report

I recommend the article to be accepted in present revised form